# Influence of pH on Molecular Hydrogen (H$_2$) Generation and Reaction Rates during Serpentinization of Peridotite and Olivine

**Ruifang Huang** [1],*, **Weidong Sun** [2,3,4],*, **Maoshuang Song** [5] and **Xing Ding** [5]

1   SUSTech Academy for Advanced Interdisciplinary Studies, Southern University of Science and Technology, Shenzhen 518055, China
2   Center of Deep Sea Research, Institute of Oceanology, Chinese Academy of Sciences, Qingdao 266071, China
3   Laboratory for Marine Mineral Resources, Qingdao National Laboratory for Marine Science and Technology, Qingdao 266237, China
4   CAS Center for Excellence in Tibetan Plateau Earth Sciences, Chinese Academy of Sciences, Beijing 100101, China
5   State Key Laboratory of Isotope Geochemistry, Guangzhou Institute of Geochemistry, Chinese Academy of Sciences, Guangzhou 510640, China; msong@gig.ac.cn (M.S.); xding@gig.ac.cn (X.D.)
*   Correspondence: huangrf@sustech.edu.cn (R.H.); weidongsun@gig.ac.cn (W.S.)

**Abstract:** Serpentinization produces molecular hydrogen (H$_2$) that is capable of supporting communities of microorganisms in hydrothermal fields, which suggests that serpentinization may be closely related to the origin of life at the early history of the Earth and possibly other planets. In this study, serpentinization experiments were performed at 300 °C and 3.0 kbar with natural olivine and peridotite as starting reactants to quantify the influence of acidic and alkaline solutions on the processes of serpentinization. The results reveal that acidic and alkaline solutions greatly influence molecular hydrogen (H$_2$) generation and the rates of serpentinization. Acidic (pH = 2.50) and alkaline solutions (pH = 13.5) increased H$_2$ production and the rates of peridotite serpentinization. Highly acidic solutions (2 M HCl), however, decreased the production of H$_2$ after peridotite serpentinization by around two orders of magnitude. The decrease in H$_2$ production was associated with a sharp decline in the rates of reaction; e.g., when peridotite was reacted with neutral solutions (0.5 M NaCl), 88% of reaction progress was achieved after an experimental duration of 27 days, and the reaction extent decreased by ~50% for experiments with highly acidic solutions (2 M HCl) over the same period. In contrast, for experiments with solely olivine, highly acidic solutions (2 M HCl) promoted the rates of olivine serpentinization and H$_2$ production. The contrasting effect of highly acidic solutions (2 M HCl) on the processes of olivine and peridotite serpentinization may reflect the influence of pyroxene minerals, which could release SiO$_2$ during peridotite serpentinization and, consequently, hydrogen generation and reaction rates may decrease. The experimental results of this study suggest that H$_2$ production and the rates of serpentinization can be greatly influenced by acidic and alkaline solutions and co-existing minerals (e.g., pyroxene).

**Keywords:** olivine; serpentinization; pyroxene minerals; aluminum; molecular hydrogen

---

## 1. Introduction

Fluids issuing from serpentinizing ultramafic rocks are typically enriched in volatiles, especially molecular hydrogen (H$_2$) and methane (CH$_4$) [1–3]. These volatiles are capable of supporting communities of microorganisms in serpentinite-hosted hydrothermal vent fields and the deep subsurface [1–3]. The microorganisms survive in relatively hot (e.g., ~80 °C), acidic and alkaline

solutions with pH ranging from <1 to 12 [1,2]. They may represent the ancestor of life in the early history of the Earth when ultramafic rocks may have been exposed more extensively in the Hadean and early Archean ocean due to higher potential temperatures compared to the present-day mantle [4]. Therefore, serpentinization may have played an important role for the origin and evolution of life in the early history of Earth and possibly other planetary bodies.

Several lines of geological and geophysical evidence suggest that peridotites of the oceanic lithosphere and mantle wedge in subduction zones are serpentinized [5–7]. Serpentinization greatly modifies the chemical and physical properties of the oceanic lithosphere [8–13]. It significantly reduces the strength of olivine [8,9], possibly leads to an increase in volume, and triggers earthquakes [5,7]. Serpentine, one of the main hydrous minerals produced after serpentinization of ultramafic rocks, contains abundant $H_2O$ (up to 13.5 wt%) and fluid-mobile elements (such as Cs, Sr, and Pb) [10–13]. As demonstrated by thermodynamic and experimental studies, serpentine can be stable at depths of greater than 150 km [14,15]. Therefore, serpentine is an important reservoir for $H_2O$ and fluid-mobile elements (such as Cs, Sr, and Pb) in subduction zones.

Despite the significance of serpentinization for the recycling of $H_2O$ and fluid-mobile elements and also for the genesis of life, only a few experimental studies have focused on the rates of reaction and molecular hydrogen ($H_2$) production [16–24]. Previous experiments were performed mostly with olivine as the starting reactant, showing that the pH of starting fluids greatly influences the kinetics of olivine serpentinization [18,19]. Hydroxyl-alkaline solutions promote the hydrothermal alteration of olivine, and acidic solutions impede serpentinization reactions [18,19]. Although olivine is one of the most abundant minerals in peridotite, it may not necessarily be an equivalent of peridotite during serpentinization. As revealed by a recent experimental study, peridotite is serpentinized at much faster rates than olivine [23]. However, the influence of pH on the kinetics of peridotite serpentinization and $H_2$ production has not been quantified, which is important for understanding the mechanisms of serpentinization processes.

In this study, we performed serpentinization experiments at 300 °C and 3.0 kbar with natural olivine and peridotite as starting reactants. The aims are to (1) study the influence of pH on the rates of serpentinization and the production of $H_2$ and (2) compare the processes of olivine and peridotite serpentinization in acid and alkaline solutions.

## 2. Materials and Methods

### 2.1. Starting Materials

Fluids that issue from hydrothermal vent fields commonly have abundant Ca (up to 66 mmol/kg), which indicates hydrothermal alteration of a lherzolite-type peridotite [2,25]. Based on these observations, the hydrothermal experiments of this study were performed with a lherzolite-type peridotite as the starting reactant. The peridotite was sampled at Panshishan (Jiangsu Province, China) where it occurs as xenoliths in alkaline basalts [26–28]. The sample is composed of ~65 vol% olivine, 20 vol% orthopyroxene, 15 vol% clinopyroxene, and 1–2 vol% spinel. More information on the peridotite can be found in previous studies [23,29,30]. The sample was disaggregated, ground in an agate mortar, and subsequently sieved into starting grain sizes of 25–50 μm. After that, they were washed in an ultrasonic bath in order to remove very fine particles.

Olivine grains were picked from crushed peridotite (>250 μm) under a binocular microscope, and those with obvious signs of weathering or inclusion of other minerals were excluded. The grains were cleaned in an ultrasonic bath to remove fine particles that had adhered to the olivine, especially pyroxene and spinel. After that, they were ground in an agate mortar and then sieved into grain sizes of 25–50 μm. Olivine grains were washed in an ultrasonic bath to remove very fine particles.

The starting fluids for the experiments of this study were neutral (0.5 M NaCl), alkaline (0.5 M NaOH, pH = 13.5), acidic (0.05 M HCl, pH = 2.50), and highly acidic (2 M HCl) solutions. Neutral and alkaline solutions were prepared with reagent-grade NaCl and deionized water (resistivity of

18.2 MΩ. cm), and acidic solutions (0.05 M HCl and 2 M HCl) were prepared by diluting concentrated HCl solutions.

## 2.2. Hydrothermal Experiments

The solid reactants and starting fluids were loaded into gold capsules (4.0 mm O.D, 3.6 mm I.D. and ~30 mm in length), and mass ratios between solid reactants and starting fluids were ~1.0. Gold has been commonly used in previous hydrothermal experiments, because gold is chemically inert and does not form any Fe–Au alloys [20,23,31]. Gold capsules were sealed by using a tungsten inert gas high-frequency pulse welder (PUK3, Lampert Werktechnik, Germany). Gold capsules were always checked for leaks before and after experiments by putting them in a drying furnace at 60 °C for around 2 h, and those without significant mass differences (<0.5%) were used in serpentinization experiments.

All of the experiments were performed in cold-seal vessels in a hydrothermal laboratory at the Guangzhou Institute of Geochemistry, Chinese Academy of Sciences (Table 1). The capsule was placed into the end of a hydrothermal vessel, followed by a filler rod (~6 cm long). Water was taken as the pressure medium, and pressure was achieved by pumping water into the vessel and measured by a pressure gauge with a precision of ±100 bar. Temperature was monitored by an external K-type thermocouple that was inserted into a hole near the end of the vessel. The accuracy of the temperature was within ±2 °C. Quenching was facilitated with immersion into water bath, and the temperature decreased to <100 °C within 10 s.

## 2.3. Analytical Methods

### 2.3.1. Gas Chromatography

Molecular hydrogen ($H_2$) produced in all of the experiments of this study was analyzed with an Agilent 7890A gas chromatograph at the State Key Laboratory of Organic Geochemistry, Guangzhou Institute of Geochemistry. The gold capsule was placed into a volume-calibrated glass pipe that was evacuated by a vacuum pump to reach the pressure of $<1 \times 10^{-2}$ Pa before measurement. The glass pipe was connected to a Toepler pump and an Agilent 7890A gas chromatograph. A blank analysis was always performed before sample measurement. After that, the capsule was pierced by a steel needle in vacuum and all gas components were concentrated and measured by gas chromatography. The gas components were quantified by using an external standard with an accuracy of <0.5%. Further details on the analytical procedures can be found in previous studies [32,33].

### 2.3.2. X-ray Diffraction

X-ray diffraction (XRD) analyses were performed with a Rigaku Smartlab 9KW diffractometer in a high-pressure laboratory at the Southern University of Science and Technology. The XRD patterns were collected using Cu k$\alpha$1 and k$\alpha$2 radiation in the range 2θ = 5–90°, with a step size of 0.02°, and a counting time of 10 s per step.

### 2.3.3. Scanning Electron Microscope

The morphology of the solid reaction products was characterized by a scanning electron microscope using a Zeiss Ultra 55 Field emission gun at Second Institute of Oceanography, State Oceanic Administration of China. Samples were dispersed on a double-sided carbon tape with platinum/carbon coatings. An accelerating voltage of 5 kV was used.

Table 1. Experimental conditions.

| Sample | T(°C) | P (kbar) | Solid Reactant [a] | Starting Solution [b] | W/R Ratios [d] | Time (days) | Srp (%) [e] | Tlc (%) | Ol (%) | Py (%) | H$_2$ (mmol/kg) |
|---|---|---|---|---|---|---|---|---|---|---|---|
| H-2 | 300 | 3.20 | Prt | 0.05 M HCl | 0.97 | 14 | 96(2.0) | - | 2.5(2.0) | 0.0 | 98 |
| H-3 | 300 | 3.20 | Prt | 2 M HCl | 0.98 | 30 | 19(1.0) | 44(2.2) | 34(1.0) | 3.0(2.0) | 1.9 |
| H-4 | 300 | 3.40 | Ol | 2 M HCl | 0.93 | 16 | 81(4.6) | - | 19(4.6) | - | 89 |
| H-5 | 300 | 3.80 | Prt | NaOH | 0.97 | 31 | 84(1.7) | - | 9.4(0.7) | 4.6(0.7) | 162 |
| H-6 | 300 | 3.45 | Prt | NaOH | 0.98 | 8 | 77(1.9) | - | 12(0.8) | 9.0(0.8) | 244 |
| H-7 | 300 | 2.30 | Prt | NaCl [c] | 0.89 | 27 | 88(3.0) | - | 9.4(1.2) | 2.6(4.2) | 181 |
| H-8 | 300 | 3.20 | Prt | 0.05 M HCl | 0.96 | 14 | 92(3.2) | - | 6.1(1.4) | 0.0 | 161 |
| H-9 | 300 | 2.79 | Ol | 2 M HCl | 0.91 | 26 | 80(1.7) | - | 20(1.7) | - | 146 |
| H-10 | 300 | 2.80 | Prt | 2 M HCl | 0.97 | 26 | 15(2.2) | 33(5.0) | 37(2.0) | 15.6(4.5) | 3.8 |
| H-11 | 300 | 3.00 | Prt | 0.05 M HCl | 1.04 | 27 | 98(2.8) | - | 2.7(2.2) | 0.0(5.0) | 124 |
| H-12 | 300 | 2.50 | Prt | 2 M HCl | 0.92 | 20 | 12(1.2) | 26(2.6) | 40(2.2) | 22(6.0) | 3.0 |
| H-13 | 300 | 3.27 | Prt | 0.05 M HCl | 0.99 | 8 | 83(1.7) | - | 10(0.3) | 5.0(0.3) | 133 |
| H-16 | 300 | 3.00 | Ol | NaCl | 1.00 | 13 | 72(6.3) | - | 28(6.0) | - | 80 |

[a] The starting grain sizes of solid reactants for all the experiments were 25–50 μm. [b] 0.05 M HCl (pH = 2.5), 0.5 M NaOH (pH = 13.5), 0.5 M NaCl (pH = 6.6). [c] 0.5 M NaCl (pH = 2.60). [d] Water/rock ratios: ratios between the mass of the starting fluid and solid reactants at the start of experiments. [e] Numbers in brackets are the standard deviation of at least three analyses (±1δ). "-" means that there was no talc or pyroxene. Ol: Olivine; Prt: peridotite; Srp: serpentine; Tlc: talc; Py: pyroxene minerals.

### 2.3.4. Fourier Transform Infrared Spectroscopy

Fourier transform infrared spectroscopy (FTIR) analyses were performed using a Bruker Vector 33 FTIR spectrometer at the State Key Laboratory of Isotope Geochemistry, Guangzhou Institute of Geochemistry. KBr pellets were prepared by mixing ~1 mg of sample powder with 200 mg of KBr. The spectra were obtained at wave numbers from 400 to 4000 $cm^{-1}$ at a resolution of 4 $cm^{-1}$, and 32 scans were accumulated for each spectrum.

## 3. Results and Discussion

### 3.1. Identification of the Solid Experimental Products

Secondary minerals were identified by XRD, SEM, and FTIR (Figures 1 and 2), and they were quantified by FTIR (Figure S1). The intensities of diffraction peaks for olivine and pyroxene minerals decreased with progressive hydrothermal alteration, and the intensities of secondary minerals (such as serpentine and talc) increased (Figure 1). Olivine and pyroxene minerals in all experiments were replaced by serpentine, except for those with highly acidic solutions (2 M HCl) and peridotite as starting reactants, where the secondary minerals were mainly composed of serpentine and talc (Figures 1 and 2). The serpentine minerals are characterized by infrared bands at 979 $cm^{-1}$, 1087 $cm^{-1}$, and 3686 $cm^{-1}$ (Figure 2) [19,34]. The bands at 979 $cm^{-1}$ and 1087 $cm^{-1}$ represent the Si–O group of serpentine and the band at 3686 $cm^{-1}$ corresponds to the –OH group [19,34]. Talc has a typical stretching mode at 671 $cm^{-1}$ for the Si–O–Mg group and a stretching vibration at 3677 $cm^{-1}$ for the –OH group [35]. The serpentine mineral type seems to depend on acidic and alkaline solutions; e.g., chrysotile was the main serpentine mineral for experiments with peridotite and alkaline solutions, and both chrysotile and lizardite were produced in experiments with peridotite and acidic solutions (Figure 2).

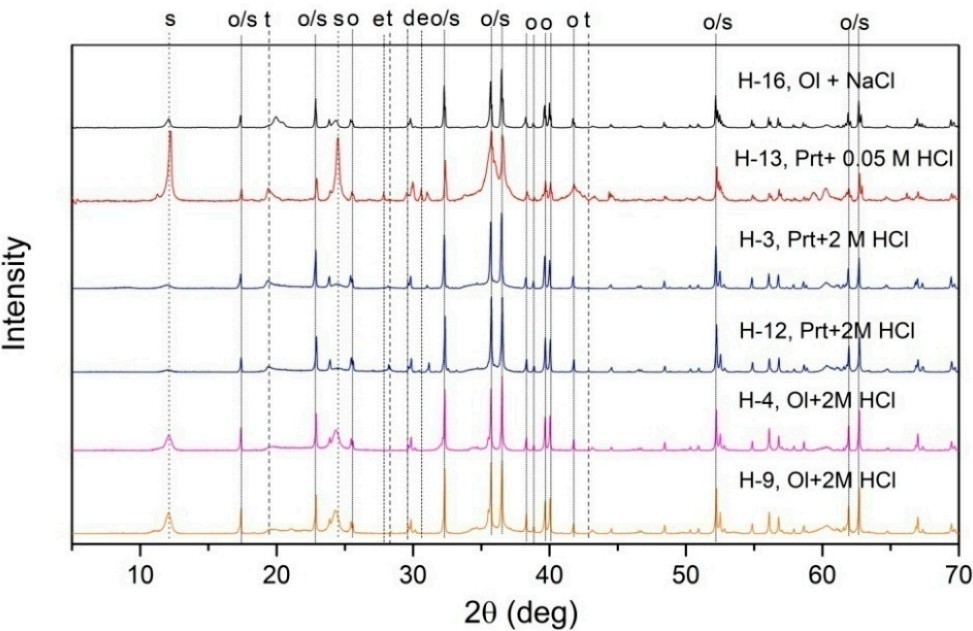

**Figure 1.** X-ray diffraction (XRD) patterns of the experimental products after serpentinization of olivine and peridotite. o: olivine; e: enstatite; d: diopside; s: serpentine; t: talc.

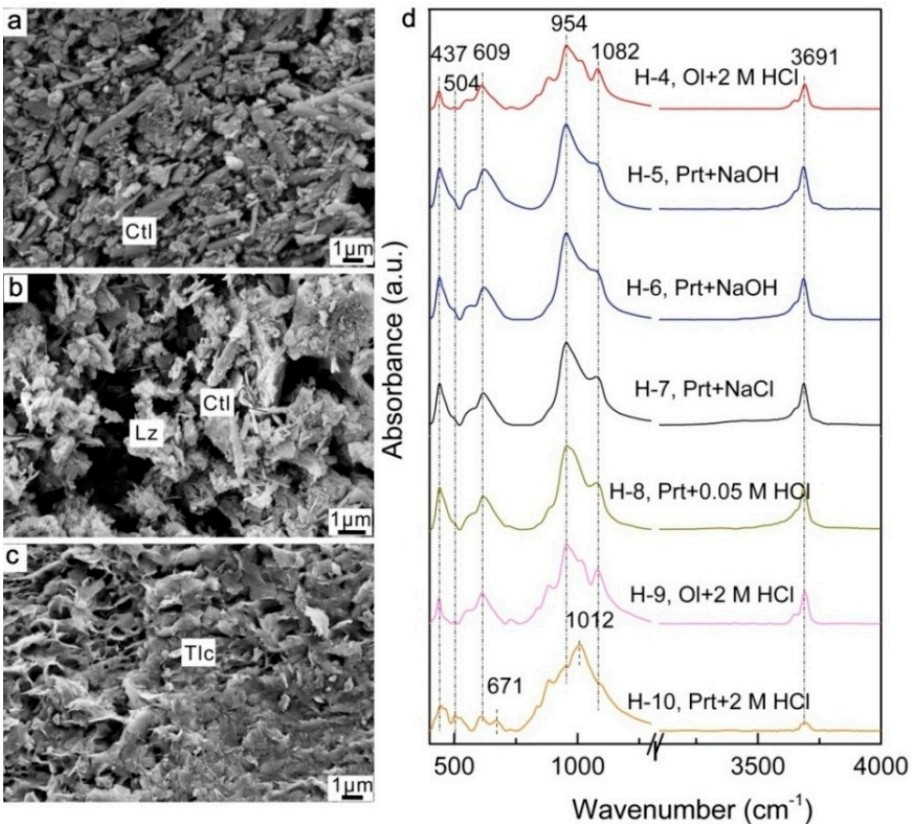

**Figure 2.** Characterization of the solid experimental products. Scanning electron microscope images of (**a**) H-6, with peridotite and alkaline solutions (0.5 M NaOH, pH = 13.5); (**b**) H-8, with peridotite and acid solutions (0.05 M HCl, pH = 2.50); (**c**) H-10, with peridotite and high acid solutions (2 M HCl) as starting reactants; and (**d**) Infrared spectra of the experimental products. Ctl: chrysotile; Lz: lizardite; Tlc: talc; Prt: peridotite.

Although brucite is predicted to form at the early stage of serpentinization by theoretical models [36], it is seldom found in oceanic and ophiolitic serpentinites [37–40]. Consistently, brucite was not detected in all of the experiments of this study. The absence of brucite can be attributed to multiple factors. First, the stability of brucite can be greatly influenced by silica activity, and high silica stability leads to brucite dissolution [37,39]. Compared to olivine, peridotite serpentinization has silica activity that is around 1–2 orders of magnitude higher [39,41], which may result in the dissolution of brucite [37,39]. Moreover, acidic and alkaline solutions may greatly influence the stability of brucite during serpentinization [42]. Previous experiments suggest that brucite is preferentially formed under highly alkaline conditions (pH > 11), and the production of brucite is impeded in acidic and neutral solutions [42]. Furthermore, it may be due to a systematically lower amount of brucite, which is difficult to identify by conventional analytical techniques.

### 3.2. Effect of Acidic and Alkaline Solutions on $H_2$ Production

Blank experiments were performed at 300 °C and 3.0 kbar by loading natural olivine and peridotite or starting fluids into gold capsules. The amounts of $H_2$ and hydrocarbons (e.g., methane, ethane, and propane) were below the detection limit of gas chromatograph even after an experimental duration of 27 days. This suggests that the starting reactants used in the experiments of this study contain no molecular hydrogen and hydrocarbons. Otherwise, hydrocarbons would decompose after heating at 300 °C, leading to the formation of short-chain hydrocarbons. Therefore, molecular hydrogen that was detected in the hydrothermal experiments was mainly produced during the reaction between olivine/peridotite and starting solutions.

Acidic and alkaline solutions were found to have a significant influence on the production of $H_2$ after peridotite serpentinization (Figure 3, Table 1). When peridotite was reacted with alkaline solutions (pH = 13.5), $H_2$ production was 244 mmol/kg after eight days of serpentinization (Figure 3), which is much higher than the amount of $H_2$ produced after serpentinization of peridotite in saline solutions (0.5 M NaCl). The decrease in $H_2$ production after eight days of serpentinization is possibly attributed to the formation of hydrocarbons (e.g., methane, ethane and propane), which were commonly produced after serpentinization [21]. Despite the decrease, $H_2$ production after 31 days of serpentinization was still higher compared to $H_2$ produced in neutral saline solutions (Figure 3). This suggests that alkaline solutions promote the production of $H_2$ after peridotite serpentinization. An increase in the production of $H_2$ was also observed for experiments with peridotite and acidic solutions (0.05 M HCl) (Figure 3). In contrast, highly acidic solutions (2 M HCl) decreased the production of $H_2$ after peridotite serpentinization by around two orders of magnitude (Figure 3, Table 1). This suggests that that the production of $H_2$ during peridotite serpentinization is not simply controlled by Reaction (1):

$$2H^+ + 2e = H_2 \tag{1}$$

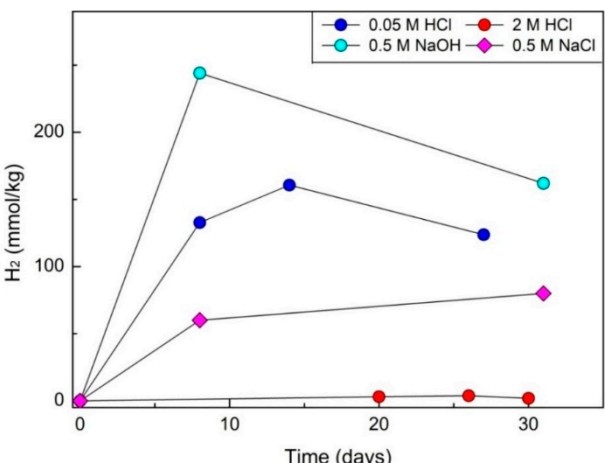

**Figure 3.** The production of $H_2$ (mmol/kg) after serpentinization of peridotite as a function of time (days).

Otherwise, highly acidic solutions would greatly enhance the production of $H_2$, which disagrees with the experimental findings of this study. Previous experimental studies have revealed that $H_2$ production is closely related to the oxidation of ferrous iron derived from olivine and pyroxene minerals into ferric iron [41,42] (Reaction (2)):

$$3Fe^{2+} + 4H_2O = Fe_3O_4 + H_2 + 6H^+ \tag{2}$$

Alkaline solutions may lead to a consumption of hydrogen ions, which possibly drives Reaction (2) to the right direction and consequently the production of $H_2$ could be enhanced. Reaction (2), however, cannot explain the increase in $H_2$ production that was observed in experiments with acidic solutions (0.05 M HCl). Taken together, these results suggest that hydrogen generation during serpentinization may not be simply controlled by Reactions (1) and (2).

A contrasting effect of highly acidic solutions (2 M HCl) on hydrogen generation during serpentinization of olivine and peridotite was observed. Highly acidic solutions (2 M HCl) promoted the production of molecular hydrogen ($H_2$) after olivine serpentinization; e.g., when olivine was reacted with saline solutions (0.5 M NaCl), 80 mmol/kg $H_2$ was produced after 23 days of reaction, and it increased to 146 mmol/kg in experiments with highly acidic solutions (2 M HCl) over a similar period (Table 1). In contrast, highly acidic solutions (2 M HCl) impeded significantly the hydrogen

production after peridotite serpentinization (Figure 3, Table 1). This contrast reflects that the presence of pyroxene and spinel in peridotite may influence the interaction between highly acidic solutions (2 M HCl) and olivine. As revealed by the X-ray diffraction and infrared spectroscopy analyses, serpentine and talc were produced during the reaction between peridotite and highly acidic solutions (2 M HCl) (Figures 1 and 2). In contrast, serpentine was the major secondary mineral for experiments with olivine and highly acidic solutions (2 M HCl). The formation of talc indicates higher silica activity due to more abundant $SiO_2$ contents of talc compared to the $SiO_2$ contents of serpentine. Thermodynamic calculations suggest that the production of $H_2$ can be greatly impeded under higher silica activity [43], which may therefore explain the great decrease in $H_2$ production that was observed for experiments with peridotite and highly acidic solutions (2 M HCl).

### 3.3. Effect of Acidic and Alkaline Solutions on the Kinetics of Serpentinization

The kinetics of serpentinization in the experiments of this study were quantified with infrared spectroscopy analyses. This technique has already been used in previous studies to determine the amounts of serpentine in soil samples and experimental products after peridotite serpentinization by calibrating the areas of the infrared hydroxyl bands [23,44]. For experiments of this study with serpentine as the major secondary mineral, the kinetics of serpentinization were quantified according to calibration curves based on mechanical mixtures of serpentine and olivine/peridotite in ratios ranging from 0 to 100% [23]. Infrared spectra of the mechanical mixtures show that the proportions of serpentine are positively correlated with integrated intensity ratios $\log(A_{3698}/A_{503})$, with higher intensity ratios for larger amounts of serpentine [23]. $A_{3698}$ is the integrated intensity of the –OH group in serpentine and $A_{503}$ is the integrated intensity of Si–O group in olivine [23]. For experiments with serpentine and talc as the main secondary minerals, the rates of reaction were determined according to a standard curve based on infrared spectra of mechanical mixtures of peridotite, serpentine, and talc (Figure S1). The amounts of serpentine and talc in the mixtures are positively correlated with integrated intensity ratios $(A_{609} + A_{670})/A_{503}$ ($R^2 = 0.97$, Figure S1), where $A_{609}$ is the integrated intensity of the Mg–O band in serpentine, $A_{670}$ is the integrated intensity of the Si–O–Mg band in talc, and $A_{503}$ is the integrated intensity of the Si–O band in olivine. The advantage of using integrated intensity ratios for calibration is that weight uncertainties can be minimized, and analyses of varied amounts of solid products yielded the same results. Repeated analyses of the same sample (>3 times) gave a precision of ±4%.

Figure 4 illustrates the extent of reaction (%) as a function of experimental duration (days), showing that pH greatly influences the kinetics of serpentinization. When peridotite was reacted with neutral solutions (0.5 M NaCl), 70% of reaction extent was achieved after an experimental duration of 19 days, and the extent of reaction increased to 88% after a longer period (27 days). In contrast, for experiments with peridotite and acidic solutions (0.05 M HCl) as starting reactants, 92% of reaction extent was achieved after 14 days, and complete serpentinization was reached after 27 days. This suggests that acidic solutions (0.05 M HCl) promote the hydrothermal alteration of peridotite. However, highly acidic solutions (2 M HCl) significantly decrease the kinetics of serpentinization; e.g., 47% of reaction extent was achieved after 26 days of hydrothermal alteration. Alkaline solutions (pH = 13.5), however, were found to have a negligible influence on the rates of peridotite serpentinization, supported by the comparable reaction extent that was achieved in experiments with alkaline and neutral solutions.

Compared to peridotite, olivine was found to have slower rates of serpentinization in neutral saline solutions. For experiments with olivine and saline solutions (0.5 M NaCl), 62% of reaction extent was achieved after 26 days of hydrothermal alteration, which is significantly lower compared to the reaction extent that was reached during serpentinization of peridotite. Such an observation agrees well with a recent experimental study [23]. Highly acidic solutions (2 M HCl), however, slightly increased the rates of olivine serpentinization (Figure 4b); e.g., 81% of reaction extent was achieved after 16 days of alteration, which is much higher than the reaction extent that was achieved in saline solutions (0.5 M

NaCl). Taken together, these results suggest that pH has a contrasting influence on the rates of olivine and peridotite serpentinization.

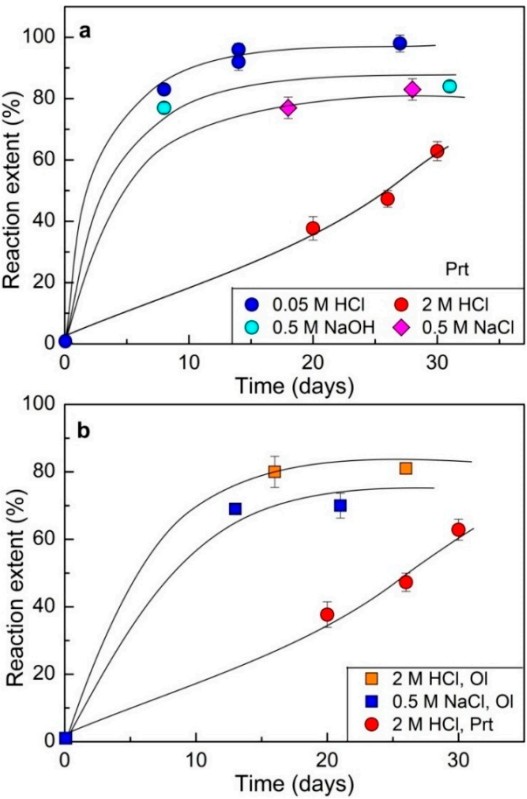

**Figure 4.** The extent of reaction (%) as a function of time (days): (**a**) shows the influence of pH on the kinetics of peridotite serpentinization; (**b**) illustrates that highly acidic solutions (2 M HCl) accelerate the rates of olivine serpentinization, and they greatly impede the hydrothermal alteration of peridotite.

The kinetic pseudo-first-order and pseudo-second-order models have been used to describe reactions at solid–fluid interfaces, e.g., sorption of water in clay minerals, crystal growth processes, and serpentinization processes [18,19,23,45]. The kinetic data of this study were successfully fitted with a pseudo-second-order model (Figure 4). This model describes a fast mass transfer followed by a slow equilibration of mass transfer in closed systems. The differential form of the kinetic model can be written as

$$\frac{d\xi}{dt} = k(\xi_{max} - \xi)^2 \tag{3}$$

where $\xi_{max}$ is the maximum reaction extent (%) achieved during serpentinization, $\xi$ is the reaction extent (%) at time $t$(day), and k is the rate constant. The integrated form of Equation (3) for the boundary conditions $t = 0$ to $t = t$ and $\xi = 0$ to $\xi = \xi$ yields a relationship between reaction extent ($\xi$) and time ($t$):

$$\xi = \frac{\xi_{max} \times t}{t_{1/2} + t} \tag{4}$$

where $t_{1/2}$ represents the time when half of the maximum reaction extent was reached. The serpentinization initial rate is described as $v_0 = \xi_{max}/t_{1/2}$.

For experiments with peridotite and highly acidic solutions (2 M HCl) as starting reactants, the kinetics of serpentinization were successfully fitted with an empirical Equation (5):

$$\xi = \xi_{max}/(1 + \exp(-\frac{t - t_{\frac{1}{2}}}{b})) \tag{5}$$

where $\xi_{max}$ is the maximum reaction extent (%) achieved during serpentinization, $\xi$ is the reaction extent (%) at any time $t$ (day), and $t_{1/2}$ is the duration when half of the maximum of the reaction extent was reached. The initial-rate of olivine serpentinization was calculated according to $v_0 = \xi_{max}/t_{1/2}$.

The kinetic parameters are summarized in Table 2. They show that pH greatly influences the initial rate of serpentinization. The initial rate of peridotite serpentinization in neutral solutions (0.5 M NaCl) is $2.10 \times 10^{-6} \cdot s^{-1}$, and it becomes around 1 order of magnitude slower in high acid solutions (2 M HCl), which suggests that highly acidic solutions decrease the initial-rate of peridotite serpentinization. Compared to peridotite, olivine has a slightly slower initial-rate of serpentinization in neutral solutions (0.5 M NaCl), $1.65 \times 10^{-6} \cdot s^{-1}$. When olivine was reacted with highly acidic solutions (2 M HCl), the initial rate of serpentinization increased to $2.46 \times 10^{-6} \cdot s^{-1}$. This suggests that highly acidic solutions speed up the hydrothermal alteration of olivine. Taken together, these results suggest that pH has a dramatically different influence on the rates of olivine and peridotite serpentinization.

**Table 2.** Kinetic parameters for serpentinization of olivine and peridotite.

| Solid Reactants | Starting Solutions | $\xi_{max}$ (%) | | $t_{1/2}$ (days) | Initial Rate ($s^{-1}$) | Fitting $R^2$ |
|---|---|---|---|---|---|---|
| | | Cal. | Exp. | | | |
| Olivine | NaCl | 100 | 70 | 7.0 ± 1.1 | $1.65 \times 10^{-6} \pm 0.26 \times 10^{-6}$ | 0.987 |
| Olivine | 2 M HCl | 100 | 81 | 4.7 ± 0.7 | $2.46 \times 10^{-6} \pm 0.37 \times 10^{-6}$ | 0.995 |
| Peridotite | NaCl | 100 | 80 | 5.5 ± 0.1 | $2.10 \times 10^{-6} \pm 0.04 \times 10^{-6}$ | 0.999 |
| Peridotite | 0.05 M HCl | 100 | 98 | 1.2 ± 0.2 | $9.35 \times 10^{-6} \pm 1.7 \times 10^{-6}$ | 0.995 |
| Peridotite | NaOH | 100 | 84 | 2.8 ± 0.8 | $4.11 \times 10^{-6} \pm 1.1 \times 10^{-6}$ | 0.984 |
| Peridotite | 2 M HCl | 100 | 63 | 25.7 ± 1.0 | $4.50 \times 10^{-7} \pm 0.18 \times 10^{-7}$ | 0.965 |

### 3.4. Mechanisms of Serpentinization Reactions

As revealed by analyses of natural serpentinites and experimental products, serpentinization processes are coupled with dissolution of minerals (olivine, orthopyroxene, and clinopyroxene) and precipitation of serpentine minerals [18,19,45,46]. The dissolution of olivine can be expressed as follows:

$$2Mg_2SiO_4 + 8H^+ = 4Mg^{2+} + 2H_4SiO_4 \tag{6}$$

Precipitation from solution and/or nucleation-growth processes produces serpentine minerals (Reaction (7)):

$$3Mg^{2+} + 2H_4SiO_4 + H_2O = Mg_3Si_2O_5(OH)_4 + 6H^+ \tag{7}$$

The summation of Reactions (6) and (7) gives a classic serpentinization reaction of olivine:

$$2Mg_2SiO_4 + 2H^+ + H_2O = Mg_3Si_2O_5(OH)_4 + Mg^{2+} \tag{8}$$

As suggested by previous experimental studies, acidic solutions with a pH ranging from 2 to 5 increase the rates of olivine and pyroxene dissolution [47–49], which may lead to an increase in the kinetics of peridotite serpentinization. Consistently, the experimental results of this study show that acidic solutions (0.05 M HCl) slightly enhance the serpentinization of peridotite (Figure 4). In contrast, highly acidic solutions (2 M HCl) decrease significantly the rates of peridotite hydrothermal alteration (Figure 4). An increase in the rates of reaction for olivine-only experiments indicates that highly acidic solutions (2 M HCl) greatly influence the hydrothermal alteration of pyroxene minerals. As suggested by previous experimental studies, the rates of clinopyroxene dissolution in acidic solutions are around 1–2 orders of magnitude faster than the rates of olivine dissolution [47–49]. Pyroxene minerals may release some of their $SiO_2$ during hydrothermal alteration, leading to the formation of talc (Figure 1). Previous experimental studies reveal that the production of a silica layer around relict olivine during serpentinization has a passivating effect on the rates of olivine serpentinization [19]. Although such a silica layer was not detected in the experimental products with conventional methods (X-ray diffraction

and FTIR) (Figure 1), it may be formed at a certain stage during serpentinization of peridotite, which greatly decreases the rates of olivine serpentinization in highly acidic solutions.

In order to illustrate the influence of acidic and alkaline solutions on the processes of peridotite serpentinization, we calibrated the proportions of serpentinized olivine, i.e., the ratios between the amounts of olivine that were replaced by serpentine and the amounts of primary olivine. The amounts of residual olivine in the experimental products were determined according to a standard curve based on infrared spectra of mechanical mixtures of peridotite, serpentine and (±) talc (Figure S2). For experiments with serpentine and talc as the main secondary minerals, the amounts of residual olivine were obtained according to a positive correlation between the amounts of olivine and integrated intensity ratios $Log(A_{503}/A_{671})$ (Figure S2); for experiments with serpentine as the major secondary hydrous mineral, the amounts of residual olivine in the run products were determined according to a standard curve based on mechanical mixtures of peridotite and serpentine [23]. The amounts of olivine that were replaced by serpentine were subsequently obtained based on mass balance. Figure 5 illustrates the proportions of reacted olivine and pyroxene (%) as a function of reaction progress (%). When peridotite was reacted with neutral solutions (0.5 M NaCl), serpentine was mainly derived from the serpentinization of olivine at the early stage of hydrothermal alteration (e.g., reaction extent of <65%) [23,42,50]. With progressive serpentinization, the proportions of serpentinized pyroxene minerals increased and they became higher than the proportions of serpentinized olivine (Figure 5). Consistently, for experiments of this study with peridotite and acidic solutions (0.05 M HCl), the proportions of serpentinized pyroxene minerals were found to be higher than the proportions of serpentinized olivine at reaction extent of >80% [23]. In contrast, for experiments with peridotite and highly acidic solutions (2 M HCl), the proportions of serpentinized pyroxene were higher than the proportions of serpentinized olivine even at the early stage of hydrothermal alteration. This suggests that highly acidic solutions (2 M HCl) accelerate the hydrothermal alteration of pyroxene minerals, which may lead to the release of more $SiO_2$ from pyroxene minerals and the formation of silica layers at a certain stage of hydrothermal alteration. Consequently, the hydrothermal alteration of olivine can be impeded greatly (Figure 5). Taken together, these results suggest that highly acidic solutions (2 M HCl) have a contrasting effect on the processes of olivine and pyroxene serpentinization, which consequently influence the rates of peridotite hydrothermal alteration.

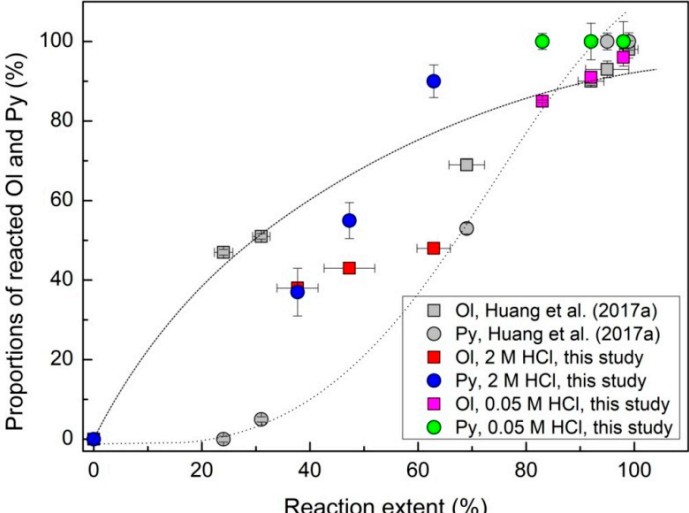

**Figure 5.** The proportions of serpentinized olivine and pyroxene minerals (%) as a function of reaction extent (%). Gray symbols are data from Huang et al. [23], and colored symbols are experimental data of the present study.

A contrasting effect of acidic and alkaline solutions on the rates of olivine and peridotite serpentinization was observed in the experiments of this study. A simple change from saline solutions (0.5 M NaCl) to highly acidic solutions (2 M HCl) increased the rates of olivine serpentinization. Highly acidic solutions (2 M HCl), however, impeded the hydrothermal alteration of peridotite serpentinization. Moreover, hydroxyl alkaline solutions enhanced the serpentinization of olivine [18], but their influence on the rates of peridotite serpentinization was much less significant (Figure 4). Such a discrepancy may be closely related to the presence of pyroxene and spinel, which released some of their aluminum during serpentinization and consequently the rates of serpentinization may be greatly influenced [38,51]. Analyses of natural serpentinites and experimental products have revealed that orthopyroxene-derived serpentine has a much lower abundance of $Al_2O_3$ compared to that of primary orthopyroxene [23,51], indicating releases of aluminum from orthopyroxene during hydrothermal alteration. It has been estimated that orthopyroxene lost ~50% of its Al during serpentinization [51]. Spinel can be hydrothermally altered, which leads to the formation of Al-depleted magnetite rinds around relict spinel [21,23]. Taken together, these results suggest that aluminum may be mobile during hydrothermal alteration of peridotite.

Indeed, aluminum is always present in serpentinites. Analyses of natural samples have revealed that serpentinites have $Al_2O_3$ contents ranging from <1.0 wt% to 20 wt% [38,51]. Olivine in natural geological settings is commonly associated with Al-bearing minerals, e.g., pyroxene, spinel, and amphibole, which can release aluminum during serpentinization. The mobility of aluminum is also supported by occurrences of metamorphic veins with abundant Al-bearing minerals such as kyanite and sillimanite [52,53]. Aluminum reacts with alkaline solutions, which consumes hydroxyl ions ($OH^-$) of alkaline solutions, and consequently alkaline solutions tend to be neutral after peridotite serpentinization. Such a hypothesis is supported by pH measurements of aqueous solutions, and it was found to be ~7.0 in experiments with alkaline solutions, consistent with previous experimental studies [18,19]. The absence of brucite in experiments of this study with alkaline solutions is a direct proof of the consumption of hydroxyl ions ($OH^-$) during peridotite serpentinization. As a consequence, the influence of alkaline solutions on the kinetics of peridotite serpentinization may be much less significant.

The contrast between the serpentinization processes of olivine and peridotite is also indicated by hydrogen generation in highly acidic solutions (2 M HCl). The experimental results of this study show that highly acidic solutions (2 M HCl) promote $H_2$ production in olivine-only experiments (Figure 3, Table 1). In contrast, highly acidic solutions (2 M HCl) decrease hydrogen generation after peridotite serpentinization by around two orders of magnitude (Figure 3, Table 1). The contrast may be attributed to the presence of pyroxene minerals, which released some of their $SiO_2$ during hydrothermal alteration of peridotite. Such an assumption is supported by the formation of serpentine and talc (Figures 1 and 2), and also by the influence of highly acidic solutions (2 M HCl) on the rates of pyroxene serpentinization (Figure 5). As suggested by previous experimental studies, the serpentinization of pyroxene releases $SiO_2$ that is around 1–2 orders of magnitude higher than $SiO_2$ leached during olivine serpentinization [41]. Thermodynamic and experimental studies have revealed that silica decreases greatly $H_2$ production after serpentinization [43]. Consistently, analyses of fluids issued from hydrothermal fields with basalts as the basement rock (e.g., Lucky Strike and TAG) suggest that $H_2$ production is very low, ranging from 0.18 mmol/kg to 0.73 mmol/kg, and fluids issued from hydrothermal fields with peridotites (such as Logachev and Rainbow) have much higher $H_2$, up to 12–16 mmol/kg [2,54]. Taken together, these results suggest that the serpentinization of olivine may be greatly influenced by pyroxene minerals. This also suggests that olivine may not be an equivalent of peridotite during serpentinization reactions.

## 3.5. Comparison with Previous Studies

The kinetics of serpentinization have been experimentally studied over the last two decades, mostly with olivine and neutral solutions (such as pure $H_2O$ and 0.5 M NaCl) as starting reactants [16,17,20,21].

Olivine is a major component of peridotite, typically >50 vol%. Analyses of natural serpentinites and experimental products have revealed that olivine serpentinization is the main serpentine-forming process at the early stage of peridotite serpentinization [23,38,42,49]. Based on these observations, olivine is proposed to be an equivalent of peridotite during serpentinization [20,21,42,49]. In a previous recent study, we performed hydrothermal experiments at 310 °C and 3.0 kbar with saline solutions (0.5 M NaCl), and demonstrated that peridotite is serpentinized at much faster rates than olivine [23]. This is attributed to the presence of pyroxene and spinel releasing aluminum that greatly accelerates the rates of olivine serpentinization [23,55]. This suggests that the process of olivine serpentinization may be distinct from the process of peridotite serpentinization. Consistently, several studies have shown that magnetite production during serpentinization of olivine and peridotite differs greatly [20,23,37,50,56]. During olivine serpentinization, magnetite has a positive correlation with reaction extent [20], and it is very low at early stages of peridotite serpentinization, but it increases abruptly at reaction extents of >70% [23,50,56]. All these observations suggest that a different mechanism exists between the serpentinization of olivine and peridotite.

Previous experimental studies suggest that serpentinization kinetics greatly depend on temperature and grain sizes of the starting reactants [16,17,20,21,42,50]. Analyses of fluids issuing from hydrothermal fields suggest that serpentinization processes can be influenced by fluid/rock ratios and fluid chemistry (including pH) [2,43,57]. To date, only a few experiments have been performed to study the influence of pH on the rates of olivine serpentinization [18,19]. They showed that alkaline solutions (pH = 13.5) promote serpentinization reactions [18], and acidic solutions (pH = 0.63) impede the serpentinization of olivine [19]. In contrast, the present study has revealed that highly acidic solutions (2 M HCl) enhance the rates of olivine serpentinization (Figure 4b). The inconsistency may be attributed to multiple factors: (1) analytical problems. Lafay et al. [19] determined the kinetics of olivine serpentinization using thermogravimetric analyses, and they proposed that infrared spectroscopy measurements yielded very large uncertainties due to the overlapping of –OH (for brucite and serpentine). However, the absence of brucite in the experimental products of the present study indicates that the influence of such overlapping is negligible. Serpentine quantified by infrared spectroscopy was positively correlated with serpentine obtained by thermogravimetric analyses ($R^2$ > 0.90) [18]. Moreover, integrated intensity ratios were used in this study for calibration to minimize weighing uncertainties, and repeated measurements on the same sample yielded a precision of ±4%. This may exclude the possibility that analytical problems are the main cause for the dramatic contrast between Lafay et al. [19] and the experimental results of this study. (2) The starting grain sizes of olivine: Lafay et al. [19] performed experiments using olivine with very small grain sizes (<30 μm), which may lead to a rapid dissolution of olivine and consequently a silica layer can be produced. As a consequence, the rates of olivine serpentinization may be greatly impeded. In contrast, the experiments of this study were performed with larger grain sizes (25–50 μm), which may not lead to a rapid dissolution of olivine, and consequently the rates of olivine serpentinization in our experiments may differ from the kinetics of olivine hydration in the experiments of Lafay et al. [19].

Molecular hydrogen ($H_2$) production during serpentinization has been experimentally studied during the last two decades, mostly with olivine and neutral solutions as starting reactants [21,36,41,57]. These studies have revealed that $H_2$ production depends on temperature, water/rock ratios, and reaction rates [21,36,41,57]. Molecular hydrogen production increases with increasing temperatures and reaches a maximum at ~300 °C, and it decreases by around 1–2 orders of magnitude at temperatures higher than 350 °C when olivine can be stable with $H_2O$ [21,36,41]. Moreover, the concentrations of $H_2$ in aqueous fluids are inversely correlated with water/rock ratios, with higher water/rock ratios for lower concentrations of $H_2$ [21]. In contrast, the production of $H_2$ has a positive correlation with serpentinization kinetics [21,42,49]. Consistently, the experimental results of this study have revealed a positive correlation between molecular hydrogen ($H_2$) production and serpentinization kinetics for experiments using olivine and highly acidic solutions (2 M HCl), and also for experiments with peridotite and acidic solutions (pH = 2.50) as starting reactants.

To date, only a few experiments have been carried out to study the influence of acidic and alkaline solutions on $H_2$ production during serpentinization [53]. High-carbonate alkalinity (1 M $NaHCO_3$) decreases $H_2$ production, which results from preferential incorporation of $Fe^{2+}$ in carbonate minerals [57]. The present study has revealed that hydroxyl alkalinity increases $H_2$ production after peridotite serpentinization, and highly acidic solutions (2 M HCl) significantly decrease $H_2$ production by around 1–2 orders of magnitude. Previous studies suggest that acidic and alkaline solutions greatly influence the stability of $Fe^{2+}$ [58–60]. Analyses of fluids issuing from hydrothermal fields have revealed that acidic fluids commonly have a higher abundance of Fe; e.g., fluids emerging from the Rainbow hydrothermal field (36°14′ N) have a pH of 2.8 and 24.0 mmol/kg Fe, and fluids issued from the Lost City hydrothermal field (30 °N MAR) have a pH of ~9.0 and Fe contents below the detection limit [59]. Analyses of fluids issued from hydrothermal fields indicate that iron is predominantly present in the form of $Fe^{2+}$ [60]. This suggests that $Fe^{2+}$ is not stable in alkaline solutions, which is preferentially hydrolyzed to form $Fe^{3+}$ (Reaction (6)), possibly leading to an increase in $H_2$ production. In contrast, highly acidic solutions (2 M HCl) may stabilize $Fe^{2+}$, and they lead to the release of more $SiO_2$ from pyroxene minerals during hydrothermal alteration of peridotite. Thermodynamic calculations have revealed that silica decreases greatly the production of $H_2$ during serpentinization [43], which may, therefore, explain the dramatic decrease in $H_2$ production that was observed in our experiments with peridotite and highly acidic solutions (2 M HCl) as starting reactants.

## 4. Conclusions

The experimental results of this study reveal that acidic and alkaline solutions greatly influence the rates of serpentinization and molecular hydrogen ($H_2$) production, and their influence on the processes of olivine and peridotite serpentinization differs greatly. Highly acidic solutions (2 M HCl) increase the rates of olivine serpentinization and the production of $H_2$. In contrast, they greatly decrease the rates of peridotite hydrothermal alteration, and $H_2$ production becomes around 1–2 orders of magnitude lower. Moreover, the influence of alkaline solutions on the rates of peridotite serpentinization was found to be negligible. In contrast, Lafay et al. [18] have revealed that alkaline solutions greatly accelerate the hydrothermal alteration of olivine. The contrast between the serpentinization of olivine and peridotite reflects the influence of pyroxene and spinel. First, pyroxene and spinel lost some of their Al during peridotite serpentinization. Aluminum accelerates the serpentinization of olivine, which may lead to an increase in $H_2$ production. Aluminum could react with alkaline solutions, resulting in a great consumption of hydroxyl ions ($OH^-$). Moreover, pyroxene minerals lost some of their $SiO_2$ during hydrothermal alteration of peridotite, especially in highly acidic solutions (2 M HCl). Releases of $SiO_2$ from pyroxene minerals may greatly decrease the production of $H_2$ [43]. In natural geological settings, fluids issuing from hydrothermal vent fields have a pH ranging from ~3.0 to 12.0 [1,2]. High-temperature fluids (e.g., ≥300 °C) are typically acidic, with very low pH, and low-temperature fluids (e.g., 80 °C) are alkaline [1,2]. Such an inverse correlation between pH of hydrothermal fluids after peridotite serpentinization and temperatures is also indicated by thermodynamic modeling [36]. Hydrothermal experiments performed at 300 °C and 500 bar show that pH of fluids (~7.6) is essentially unchangeable during serpentinization [59]. This suggests that peridotites in natural geological settings may extensively interact with acidic vent fluids. Therefore, experimental results of this study are important for understanding the processes of serpentinization, especially for serpentinization in acidic fluids. However, additional experiments on the influence of alkaline fluids at relatively low temperatures (e.g., ≤200 °C) are required. Previous experiments conducted at 200 °C and 500 bar suggest a dramatic increase in pH with increasing reaction time, initially from 6.2 to 12.2 after 656 hours [42], suggesting that alkaline fluids may greatly influence the serpentinization of peridotite at low temperatures (e.g., ≤200 °C).

**Supplementary Materials:** The following are available online at http://www.mdpi.com/2075-163X/9/11/661/s1, Figure S1: (a) A standard curve for quantifying serpentine in experiments with peridotite and high acid solutions (2 M HCl) as starting reactants. (b) A standard curve for quantifying talc in experiments with peridotite and high

acid solutions (2 M HCl) as starting reactants. Figure S2: A standard curve for obtaining the amounts of residual olivine in experiments with peridotite and high acid solutions (2 M HCl) as starting reactants.

**Author Contributions:** R.H. conceived of the primary idea, conducted all of the experiments, analyzed most samples and wrote the manuscript. W.S. and M.S. co-write the manuscript. X.D. and M.S. analyzed some samples. All authors participated into discussions and manuscript revisions.

**Funding:** This work was financially supported by the Natural Science Foundation of China (41873069), the National Key R&D Program of China (2016YFC0600408), the Strategic Priority Research Program of the Chinese Academy of Sciences (XDA22050103), and Shenzhen Clean Energy Research Institute (CERI-KY-2019-003).

**Acknowledgments:** We thank Jihao Zhu from the Second Institute of Oceanography, State Oceanic Administration of China for performing scanning electron microscope imaging. Thanks also to Shan Jiang from South China University of Technology for the help during FTIR analyses.

**Conflicts of Interest:** The funders had no role in the design of the study; in the collection, analyses, or interpretation of data; in the writing of the manuscript, or in the decision to publish the results.

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
