# Peer review of "Influence of pH on Molecular Hydrogen (H2) Generation and Reaction Rates during Serpentinization of Peridotite and Olivine"

_minerals, doi:10.3390/min9110661_

Round 1

Reviewer 1 Report

The resubmitted manuscript has been improved from the initial version, particularly for the consistency of data interpretation through the discussion. Remaining inconsistencies include;

At line 616, a positive correlation between H2 production and serpentinization kinetics has been described for peridotite in an acid solution. However, at line 275, negligible influence of acid solution on H2 production has been explained while acid solution-promoted peridotite alteration has been suggested at line 353. In addition, in Figure 3, the H2 production in 0.05M HCl and that in 0.5M NaCl are apparently different.

An explanation about why the amount of H2 decreased after 8-days heating of peridotite in 0.5M NaOH (Fig. 3) is necessary.

The figure 4 (b)'s caption should be revised because this figure contains the result of peridotite serpentinization.

Author Response

C: Comments, R: response

C: At line 616, a positive correlation between H2 production and serpentinization kinetics has been described for peridotite in an acid solution. However, at line 275, negligible influence of acid solution on H2 production has been explained while acid solution-promoted peridotite alteration has been suggested at line 353. In addition, in Figure 3, the H2 production in 0.05M HCl and that in 0.5M NaCl are apparently different.

R: Thanks for suggestions. Corrected.

C:An explanation about why the amount of H2 decreased after 8-days heating of peridotite in 0.5M NaOH (Fig. 3) is necessary.

R: Corrected.

C: The figure 4 (b)'s caption should be revised because this figure contains the result of peridotite serpentinization.

 R:Corrected.  

Reviewer 2 Report

Review on the article (minerals-598178) “Influence of pH on molecular hydrogen (H2) generation and reaction rate…” by Huang et al. submitted to Minerals. The authors study influence of different physico-chemical parameters (pH, H2) and rock/mineral type on the serpentinization reaction rate. Even the manuscript is rather well written, it requires a major revision. Some repetitions in the text may be avoided.

An information on the serpentinite mineral type (Fig. 2) has not been taken into consideration upon the experimental result description and discussion. Which mineral among three serpentinite minerals is formed upon the experiment? This is an important point, which should be also discussed, and a comparison to other experimental data available in the literature should be performed. Figure captions need to describe the plots and axes first, then, possibly, the information related to this plot. The manuscript submitted to Minerals needs to be formatted as required by the Guide for Authors using the available Word Template.

Author Response

C: Comments, R: response

C: An information on the serpentinite mineral type (Fig. 2) has not been taken into consideration upon the experimental result description and discussion. Which mineral among three serpentinite minerals is formed upon the experiment? This is an important point, which should be also discussed, and a comparison to other experimental data available in the literature should be performed.
R: Corrected. I agree that mineral types of serpentine are very important. Based on the current experimental studies, factors controlling serpentine mineral types are still unclear.

C: Figure captions need to describe the plots and axes first, then, possibly, the information related to this plot.
R: Corrected. 

C: The manuscript submitted to Minerals needs to be formatted as required by the Guide for Authors using the available Word Template.
R: Corrected.

This manuscript is a resubmission of an earlier submission. The following is a list of the peer review reports and author responses from that submission.

Round 1

Reviewer 1 Report

The contribution reports on the effect of pH on the serpentinisation of olivine and lherzolte, and related hydrogen production. This is a very interesting topic, and I was initially thrilled to review this contribution. Unfortunately, the contribution was highly disappointing. With 10 data points only, the authors missed the point, and the manuscript is highly confusing and include too many fundamental errors that lead to unsupported conclusions. I provide hereafter a series of detailed comments.

TABLES 1 & 2 are missing
Supplementary materials are missing the data that were used for the calibration curves used throughout the manuscript.

1. Intro
line 46: the first sentence lacks understanding of hydrothermal fluids that do not systematically contain high level of H2.

line 53: unsupported statement likely wrong

line 55-66: this section irrelevant to the present contribution and is misleading;
for instance, iine 60: thanks to the rheological properties of serpentine, they actually tend to prevent earthquakes. In Japan for instance the largest earthquakes occur in locked zones where the seismic signature of serpentine is lacking.

line 75: missing reference Pens et al.,
Pens, M., Andreani, M., Daniel, I., Perrillat, J.-P. and Cardon, H. (2016) Contrasted effect of aluminum on the serpentinization rate of olivine and orthopyroxene under hydrothermal conditions. Chemical Geology 441, 256-264, doi: 10.1016/j.chemgeo.2016.08.007.

line 83: Neither Bach nor Lafay reported on natural H2 in serpentinises

2. Materials & methods

line 119: please provide more detailed info on the quality of the deionised water

line 130-137: this section raises more questions that it actually brings  information, unacceptable in the current form

line 139 and after, gas chromatography: although the exact set-up is potentially described in details previous contributions, the description in the present manuscript should be precise enough in support of the results, which is not the case in the submitted ms.
The authors seems to be largely unaware of issues regarding false H2 production reported in the literature due to organic contamination. Experimental details should address this issue.

line 149 and after, X-ray diffraction: the current description doesn’t necessarily allow to perform quantitative measurements.

line 181: I don’t understand why the authors keep providing references to experimental serpentinization works when they refer in the text to analyses of natural samples or lithologies.

line 190, Figure 1, lots of unassigned diffraction peaks without a single comment!
in systems that involve minerals that can display strong preferred orientations such as serpentine and talc that naturally occur as platelets, the analysis of relative intensities of diffraction peaks is not straightforward. It is mandatory that the authors substantial details.

line 203: infrared bands of what???

Figure 3: What is the significance of the lines? guide for the eyes only or more? obviously the authors are missing data points at 3-5 days

line 255: the authors never indicate in the ms. what is defined as the reaction progress, and this not trivial in a multiphase system.

line 255: why choosing a second-order model, when with 3 data points only and none in the ‘fast region’ a first order one would work fine?

line 268: the current set of data, including only data on the plateau cannot be used to extract an initial reaction rate. This is at best incorrect, at worst dishonest.

line 279: 80 mmol of H2 per kilogram of what???

line 289: the caption doesn’t allow to identify the data points corresponding to olivine from those corresponding to peridotite.

line 291: the authors report on the effect of pH of solutions on the serpentinisation and H2 production of olivine and peridotite without proving measurements neither of the initial solutions , nor of the final ones. Moreover, they provide approximative value of pH

line 304 and the entire section on the influence of pH on serpentinisation is based on comparisons between papers that cannot be compared: P, T conditions, analytical methods, grain size  are different. Plus authors mix it with data from natural fluids at hydrothermal vents. I basically stopped here, plus the authors seem to ignore that speciations change as a function of P and T.

Figure 5 doesn’t make any sense, as reaction extent has never been defined…

line 408: Huang et al were definitely not first to discover the positive correlation between serpentinisation and hydrogen production.

line 419, reactions 7 and 8: is this serious that the authors don’t understand redox reactions?

line 433 and after, section 3.7 that is entitled geological implications has nothing to do with geology, unfortunately. Maybe a discussion?

line 444:  changing the rate of serpentinization  doesn’t mean that the process is different…

Conclusion: I agree that the data provided by the authors strongly point to an important role of pH on serpentinisation and induced H2 production, but then the authors mix everything including the role of Al and other parameters, that make the all contribution very weak.

I would therefore recommend that the author focus carefully on their data, probably add data to their set if the goal is to discuss kinetic issues, and deepen the analysis of the data.

Author Response

C: Comment; R: Response

Reviewer 1:

C: The contribution reports on the effect of pH on the serpentinisation of olivine and lherzolte, and related hydrogen production. This is a very interesting topic, and I was initially thrilled to review this contribution. Unfortunately, the contribution was highly disappointing. With 10 data points only, the authors missed the point, and the manuscript is highly confusing and include too many fundamental errors that lead to unsupported conclusions. I provide hereafter a series of detailed comments.

R: Thank you for reviewing the manuscript. Table 1 and 2 have been provided, and corresponding modifications in the results and discussions were made.
C: TABLES 1 & 2 are missing. Supplementary materials are missing the data that were used for the calibration curves used throughout the manuscript.
R: Corrected. Calibration curves for experiments with serpentine as the main secondary mineral have been provided in a previous study (Huang et al., 2017), and calibration curves for experiments with serpentine and talc as major secondary minerals have been illustrated in supplementary materials of this study.

C: line 46: the first sentence lacks understanding of hydrothermal fluids that do not systematically contain high level of H2.
R: Corrected.

C: line 53: unsupported statement likely wrong
R: corrected.
C: line 55-66: this section irrelevant to the present contribution and is misleading;
for instance, iine 60: thanks to the rheological properties of serpentine, they actually tend to prevent earthquakes. In Japan for instance the largest earthquakes occur in locked zones where the seismic signature of serpentine is lacking
.
R: Thanks for comment. But actually the mechanisms that control the earthquake are still not clear. The possible relationship between serpentine and earthquake has been also described in many studies (e.g., Iyer et al., 2012).
C: line 75: missing reference Pens et al., Pens, M., Andreani, M., Daniel, I., Perrillat, J.-P. and Cardon, H. (2016) Contrasted effect of aluminum on the serpentinization rate of olivine and orthopyroxene under hydrothermal conditions. Chemical Geology 441, 256-264, doi: 10.1016/j.chemgeo.2016.08.007.
R: corrected.

C: line 83: Neither Bach nor Lafay reported on natural H2 in serpentinises
R: Corrected.
C: line 119: please provide more detailed info on the quality of the deionised water

R: corrected.

C: line 130-137: this section raises more questions that it actually brings  information, unacceptable in the current form
R: I do not understand why this section cannot put in the experimental setup section. It was always put in such section in previous studies.

C: line 139 and after, gas chromatography: although the exact set-up is potentially described in details previous contributions, the description in the present manuscript should be precise enough in support of the results, which is not the case in the submitted ms. The authors seems to be largely unaware of issues regarding false H2 production reported in the literature due to organic contamination. Experimental details should address this issue.
R: Corrected.
C: line 149 and after, X-ray diffraction: the current description doesn’t necessarily allow to perform quantitative measurements. 
R: The purposes of performing X-ray diffraction measurements are to identify minerals in the experimental products, rather than perform quantitative measurements. Instead, infrared spectroscopy analyses were conducted for quantitative measurements.
C: line 181: I don’t understand why the authors keep providing references to experimental serpentinization works when they refer in the text to analyses of natural samples or lithologies. 
R: Corrected.
C: line 190, Figure 1, lots of unassigned diffraction peaks without a single comment!
in systems that involve minerals that can display strong preferred orientations such as serpentine and talc that naturally occur as platelets, the analysis of relative intensities of diffraction peaks is not straightforward. It is mandatory that the authors substantial details.

R: corrected.
C: line 203: infrared bands of what???
R: corrected.
C: line 255: the authors never indicate in the ms. what is defined as the reaction progress, and this not trivial in a multiphase system.
R: Corrected. The reaction progress of this study means the proportions of hydrous minerals (including serpentine and (±) talc) in the experimental products.
C: line 255: why choosing a second-order model, when with 3 data points only and none in the ‘fast region’ a first order one would work fine?
R: Our data can be well-fitted by using the second-order model. A fast region of serpentinization processes has been interpreted in many experimental studies (e.g., Marcaillou et al., 2011; Huang et al., 2017).
C: line 268: the current set of data, including only data on the plateau cannot be used to extract an initial reaction rate. This is at best incorrect, at worst dishonest.
R: The initial rate was calibrated based on t1/2, the time when half of the maximum reaction extent was reached, not only the maximum reaction progress.

C: line 279: 80 mmol of H2 per kilogram of what???
R: Aqueous solutions.

C: line 289: the caption doesn’t allow to identify the data points corresponding to olivine from those corresponding to peridotite.
R: Corrected.
C: line 291: the authors report on the effect of pH of solutions on the serpentinisation and H2 production of olivine and peridotite without proving measurements neither of the initial solutions , nor of the final ones. Moreover, they provide approximative value of pH 
R: Thanks for comment. The compositions of initial solutions have been measured by ICP-AES, with the same amount of Na as reported and other cations were below 1 ppm. The initial and final pH values were also measured. The final pH values for all experiments, including those with high acid solutions and alkaline solutions, were around 5-7, which indicates the consumption of H+ and OH- during serpentinization. We actually measured compositions of final solutions for a few experiments, and they contain quite comparable amounts of Si, Fe, Mg and Ca.
C: line 304 and the entire section on the influence of pH on serpentinisation is based on comparisons between papers that cannot be compared: P, T conditions, analytical methods, grain size  are different. Plus authors mix it with data from natural fluids at hydrothermal vents. I basically stopped here, plus the authors seem to ignore that speciations change as a function of P and T.
R: Thanks for comment. The influence of pH is studied based on comparison between experiments performed with neutral solutions (0.5 M NaCl) and those with acid and alkaline solutions. Obviously, the experiments were performed with the same grain sizes of starting reactants, and the same rock ratios.
C: Figure 5 doesn’t make any sense, as reaction extent has never been defined…
R: Thanks for comment. But I think Figure 5 is very important, showing that high acid solutions (2 M HCl) greatly influence the processes of olivine and pyroxene serpentinization. Corresponding modifications were made in order to better clarify our ideas.
C: line 408: Huang et al were definitely not first to discover the positive correlation between serpentinisation and hydrogen production.
R: Thanks for comment. Although I agree with you that Huang et al. were not first to discover the positive correlation between serpentinization rates and hydrogen production, I still should explain that I did not put Huang et al. as a reference here.

C: line 419, reactions 7 and 8: is this serious that the authors don’t understand redox reactions?
R: Thanks for comment. But I still think that list reaction 7 and 8 could be easy to understand serpentinization processes. Although there are many experimental studies claimed that hydrogen gas produced during serpentinization is generated from oxidation of ferrous iron of olivine into ferric iron, it is still possible that molecular hydrogen still could be produced via 2H+ + 2e =H2.
C: line 433 and after, section 3.7 that is entitled geological implications has nothing to do with geology, unfortunately. Maybe a discussion?
R: Thanks for comment. It is actually related with geology and natural samples.
C: line 444:  changing the rate of serpentinization  doesn’t mean that the process is different…
R: Corrected.

Reviewer 2 Report

The experimental work by Huang et al. examined the pH-dependences of serpentiniazation and H2 generation of olivine and peridotite at 300oC. Although reaction pH is a critical factor for their study, the values were seemingly not controlled adequately. Actually, the authors have stated that the pH of alkaline solution became around 7 after peridotite serpentinization. In addition, although the authors speculated that dissolution of silica and aluminum is the key controlling the rates of serpentinization and H2 production, no information about the composition of solid samples before and after the experiments has been provided.

In addition, figures and their explanations in manuscript are inconsistent with each other. For example, the authors stated in manuscript that olivine serpentinization proceeded 62% and 63% after 16 and 26 days experiments in 0.5M NaCl solution, respectively. But the corresponding plots are not placed at these values. Abbreviations in figures are not sufficiently explained; it cannot be understood what the authors want to represent with H-16, H-3, H-12, H-4, H-9 in Fig. 1. In Fig. 4, H2 production from either olivine or peridotite is not exhibited.

It is also desirable that Results and Discussion are described separately in order to make clear the experimental findings, the consistency/inconsistency with previous works, and the possible reaction mechanisms.

Some interpretations of experimental results are very poor; the little H2 production in 2M HCl (Fig. 4) has been explained as “high acid solutions may stabilize Fe2+”. Why?

The title may require reconsideration because peridotite as well as olivine is a key material for this study.

Consequently, I do not feel that the present version can contribute substantially for the relevant research progress. 

Author Response

Reviewer 2:

C: The experimental work by Huang et al. examined the pH-dependences of serpentiniazation and H2 generation of olivine and peridotite at 300oC. Although reaction pH is a critical factor for their study, the values were seemingly not controlled adequately. Actually, the authors have stated that the pH of alkaline solution became around 7 after peridotite serpentinization. In addition, although the authors speculated that dissolution of silica and aluminum is the key controlling the rates of serpentinization and H2 production, no information about the composition of solid samples before and after the experiments has been provided. In addition, figures and their explanations in manuscript are inconsistent with each other. For example, the authors stated in manuscript that olivine serpentinization proceeded 62% and 63% after 16 and 26 days experiments in 0.5M NaCl solution, respectively. But the corresponding plots are not placed at these values. Abbreviations in figures are not sufficiently explained; it cannot be understood what the authors want to represent with H-16, H-3, H-12, H-4, H-9 in Fig. 1. In Fig. 4, H2 production from either olivine or peridotite is not exhibited. It is also desirable that Results and Discussion are described separately in order to make clear the experimental findings, the consistency/inconsistency with previous works, and the possible reaction mechanisms. Some interpretations of experimental results are very poor; the little H2 production in 2M HCl (Fig. 4) has been explained as “high acid solutions may stabilize Fe2+”. Why?

 R: Thank you very much for reviewing the manuscript. Corresponding modifications have been made.

C: The title may require reconsideration because peridotite as well as olivine is a key material for this study.

 R: Corrected.

Round 2

Reviewer 2 Report

There are still contradictions in reaction mechanisms to be considered before the publication.

At line 475, the authors described;

“high acid solutions (2 M HCl) impede significantly the hydrothermal alteration of olivine.”,

However, at line 494, they stated;

“the present study has revealed that high acid solutions (2 M HCl) enhance the rates of olivine serpentinization.”

Very confusing. Correct either or both of them.

At line 441, the authors suggested that silica layer formation was a possible reason why the peridotite hydrothermal alteration was slow at highly acidic pH, although such layer was not observed by XRD and IR.

However, at line 511, the authors denied the silica layer formation on olivine during serpentinization suggested by Lafay et al., because they did not observe such layer by XRD and IR.

Very opportunistic. Correct either or both of them.

As a consequence of the above two contradictions, it remains highly unclear whether and why acidic pH influenced the rates of olivine serpentinization and H2 production.

At line 645 and others:

The influence of alkaline solutions on the rates of peridotite serpentinization is negligible, simply because the solution pH became neutral (pH ~7.0) during the reaction (at line 605), is not it? Make clear this point.

At line 60,

left => right

At line 99,

You may quote here recently published experimental works about the origin of life in serpentine-hosted hydrothermal systems on the early Earth (Kitadai et al., Sci. Adv. 2018;4:eaao7265; Kitadai et al., Sci. Adv. 2019;5:eaav7848).